# ARIAS: An AR-based interactive advertising system

**Qiujiao Wang** [1]*, **Zhijie Xie** [2]

**1** Department of Foundation, Southwest Jiaotong University Hope College, Chengdu, China, **2** Shanghai Jiangda Technology Development Co., Ltd, Shanghai, China

* wantchoosejoy@163.com

**Data Availability Statement:** All source code and data set are available at https://dx.doi.org/10.17504/protocols.io.e6nvwjyq2lmk/v2.

**Funding:** The authors received no specific funding for this work.

## Abstract

In this paper, we present an interactive advertising system based on augmented reality(AR) called ARIAS, which is manipulated with gestures for displaying advertising videos. Two-dimensional markers are defined in the system. The system captures the frame data through the camera in real time, uses OpenCV library to identify the predefined markers, and calculates the pose of markers captured by the camera. With OpenGL library, a virtual cubic model is created at the position of the marker, and thus videos or images are displayed on the six faces of the cube. The virtual cube, together with the original frame data collected by the camera, is displayed in the interactive window to achieve the augmented reality effect. Customers are accessible to various advertising content by observing the marker from different positions. The system, meanwhile, supports gesture operation in order to make the customers pay attention to the content they are interested in with one hand. The MediaPipe Hand framework is used to extract the landmarks of hands, based on which, a series of gestures are designed for interactive operation. The efficiency and accuracy of the system are tested and analyzed with the result, indicating that the system has high reliability and good interactiveness. This system is open at https://github.com/wanzhuxie/ARIAS/tree/PLOS-ONE.

## Introduction

Augmented reality (AR) is a technology based on graphics and image processing. It can overlay rich virtual objects in the real scene, making the description of real things more intuitive, detailed and interesting [1, 2]. People's cognitive speed and acceptance of real things will be improved by using AR technology [3, 4]. The practical application of augmented reality began in 1968, and has developed rapidly in recent years. There are relatively mature research results in many fields, such as education [5–7], medicine [8–10], man-machine cooperation [11–13], venue experience [14–17] and advertising [18, 19].

The AR marketing revenue in the United States is expected to reach $8.02 billion by 2024. AR advertising contains more human emotions [20], improving the relationship between customers and brands, and then, engaging consumers' purchase more frequently and rendering them more passionate about brands [21]. Marketers embrace AR technology for its increased visual attention and curiosity, enhanced memory encoding as compared to non-AR equivalent [22]. Interactive modalities, such as gestures and body positions, can improve users' sense of

**Competing interests:** The authors have declared that no competing interests exist.

self-presence and psychological engagement more strongly, and then, positively affect their satisfaction, purchase intention and memory [23].

This paper proposes an advertising video display system based on augmented reality. The plane of video display is not the normal screen plane, but the faces of a virtual cube. The faces are created from a planar marker collected by the camera, and have certain angles with the screen plane. When the angle of the marker changes, the angle of the display video relative to the display plane will change correspondingly in real time; when multiple cameras collect the marker from different directions at the same time, the displaying videos will be observed differently by customers. In short, customers can see different advertising content by observing the marker from different positions. At the same time, we have added the function of gesture operation to support users to front, move, rotate and zoom in or out the virtual cube with only one hand.

## Foundations

### External library

We use several excellent open source program libraries and frameworks to realize the design of the system and the algorithm, such as OpenCV, OpenGL, QT and MediaPipe. By virtue of these libraries and frameworks, we do not need to implement the algorithms one by one, which makes the implementation of the algorithm more convenient.

### Marker design

A marker is the foundation for marker-based augmented reality system and this system finally creates virtual object on the marker. The marker can be determined according to the actual requirements, which can be a text [11], a specific image [12], a face image [13], etc. For different markers, corresponding recognition algorithms can be designed. In this system, white cells with 5 rows and 5 columns on a black background are used to establish the marker. As shown in Fig 1, the position of white cells represents the number 1 and the others represent the number 0. In fact, the content of the marker can be regarded as coding information. By identifying the image color corresponding to each cell block, the information of each cell is identified, and the matrix information of the whole marker will also be decoded immediately. For easier recognition, we use a black border to surround the marker content.

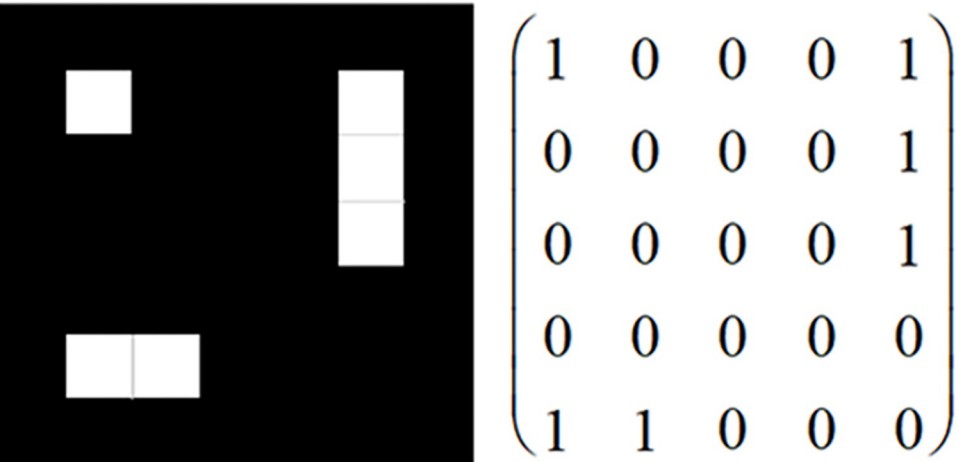

**Fig 1. Matrix of markers and information.**

The appropriate marker has the following preconditions:

1. In order to eliminate the rotational symmetry of the marker and ensure the uniqueness of the matrix information, the marker has one and only one isolated cell among its four corner cells, indicating that all the cells surrounding the isolated cell are black. The marker with the isolated cell in the upper left corner is regarded as a standard marker. One marker is considered the result of the planar rotation of the standard marker when the isolated cell is placed at the other three corners.

2. In order to avoid errors while recognizing the whole marker area, one white cell at least is requested in the last row and column. It means that the sum value of the last row and column cannot be 0.

## Camera calibration

When calculating the pose matrix of the marker mapped from the three-dimensional space to the two-dimensional image, it is necessary to input camera parameters, which can be obtained through camera calibration [24, 25]. Chessboard camera calibration is one of the most popular calibration methods for plane information acquisition and it is of high accuracy. This system uses the chessboard method for camera calibration.

## System design

A video is composed of a sequence of images, by which, generally, we mean frames. The system obtains the frames of the camera in real time, and detects whether there are predefined markers in the images. If a marker is found, the posture of the marker will be calculated by the parameters of the marker's corner coordinates in the 3D world coordinate system and the corresponding coordinates in the 2D image coordinate system. Then the coordinate system of the current OpenGL model view will be set and a virtual cube in the coordinate system will be created, on whose faces, the frames of the advertising video will be set immediately. By performing the above operations repeatedly, the frames are continuously transformed to achieve the purpose of displaying the videos. The detailed design is given in following chapters. This system supports the recognition of multiple markers, and only one of them will be presented and explained.

## Major processes

**Step 1** Basic data preparation

1. Camera calibration: Obtain the internal parameters of the camera through the referred camera calibration method.

2. Marker size setting: For the subsequent perspective transformation and marker recognition, it is necessary at first to clarify the size of the marker and the coordinate matrix $M_c$ of the four corners of the marker in the real world coordinate system. The coordinate matrix $M_c$ can be expressed as

$$M_{ci} = (X_{ci}, Y_{ci}, Z_{ci}, 1)^T \tag{1}$$

where $i$ is the index of the four corners in the real world coordinate system. $X_{ci}$, $Y_{ci}$ and $Z_{ci}$ are the coordinate values of each corner.

It is necessary for the perspective transformation and extracting the image area where the marker is located after the transformation. The corner coordinates of the marker are related to

the side length of the white cell and the position of the marker. In this paper, the side length of the white cell $L_C$ is set to 10, and the upper left corner is set at the origin. Therefore, the side length of the marker is

$$L_{\mathrm{M}} = 5L_C \tag{2}$$

And the coordinates of the marker's corners can be set to four points: upper left (0,0), lower left (0,50), lower right (50,50) and upper right (50,0).

**Step 2** Calculate the coordinate matrix $M_{\mathrm{w}}$ corresponding to the marker's corners in the image coordinate system. The coordinate matrix $M_{\mathrm{w}}$ can be expressed as

$$M_{\mathrm{w}i} = (\mathrm{X}_{\mathrm{w}i}, \mathrm{Y}_{\mathrm{w}i}, \mathrm{Z}_{\mathrm{w}i}, 1)^{\mathrm{T}} \tag{3}$$

where $i$ is the index of the four corners in the coordinate system. $\mathrm{X}_{\mathrm{w}i}$, $\mathrm{Y}_{\mathrm{w}i}$ and $\mathrm{Z}_{\mathrm{w}i}$ are the coordinate values of each corner.

Details of this step will be elaborated in the *Marker image recognition* Section.

**Step 3** Calculate the rotation and translation parameters transformed from $M_{\mathrm{c}}$ to $M_{\mathrm{w}}$. They meet the following simplified relationship

$$M_{\mathrm{w}} = M_{\mathrm{T}}M_{\mathrm{c}} \tag{4}$$

$$M_{\mathrm{T}} = \begin{pmatrix} R & T \\ 0 & 1 \end{pmatrix} \tag{5}$$

where $M_{\mathrm{T}}$ is the transform matrix, $R$ is the 3×3 rotation matrix to be solved, $T$ is 3×1 translation matrix to be solved.

In this paper, the OpenCV library function *cv::SolvePnP* is used to calculate the $R$ and $T$ combined with the camera internal parameters. The model view transformation matrix should be further calculated according to the matrix format of OpenGL.

**Step 4** Set the projection matrix of OpenGL according to the projection relationship. There are two ways of projection: perspective projection and orthogonal projection, and this system adopts the perspective projection. When the scene is symmetrical, its projection matrix is

$$\begin{pmatrix} \dfrac{2n}{w} & 0 & 0 & 0 \\ 0 & \dfrac{2n}{h} & 0 & 0 \\ 0 & 0 & -\dfrac{f+n}{f-n} & -\dfrac{2fn}{f-n} \\ 0 & 0 & -1 & 0 \end{pmatrix} \tag{6}$$

As shown in Fig 2, where

$w$ is the width near the clipping face

$h$ is the height near the clipping face

$n$ is the distance between the near clipping plane and the camera

$f$ is the distance between the near clipping plane and the camera

**Step 5** Set the projection matrix and model view matrix of OpenGL calculated in the step 3 and step 4 respectively.

**Step 6** Create the virtual cube. The local coordinate system of the model is now in effect after the step 5. Draw the virtual cube with OpenGL functions under the current coordinate system, and the side length of the cube is equal to the marker's side length $L_{\mathrm{M}}$.

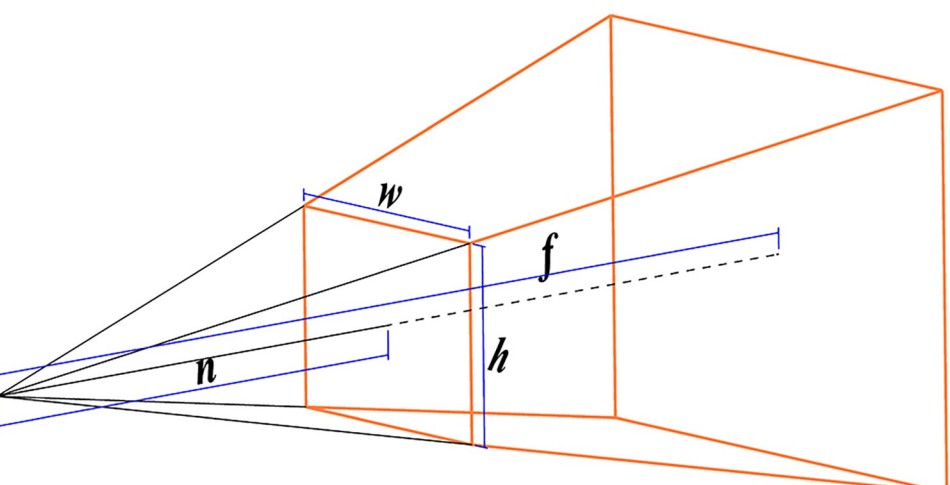

**Fig 2. Schematic diagram of a visual body.**

**Step 7** Set image texture for each face of the virtual cube. The form of videos can be set as needed and the design method will be detailed in the ***Texture Creation*** Section.

The major flow of the system is shown as Fig 3. After the above series of processing, the virtual cube will be attached to the marker and displayed together with the original frame of the video, as shown in Fig 4. The top face of the cube shows advertising video while the four vertical faces show static images.

## Marker image recognition

Before identifying different markers, we need to design the corresponding recognition algorithm according to the characteristics of the markers. For the markers referred in this paper, the recognition process is as follows:

**Step 1** Gray the original image. Graying operation is frequently used image processing because it can reduce the image memory.

**Step 2** The image should be binarized to highlight the target area. The difference between the foreground and background of the image may be great when collecting markers information in different scenes. Generally, using the same threshold globally cannot distinguish the foreground well from the background, because of which, it is necessary to use an adaptive threshold for binarization.

The adaptive threshold algorithm calculates the local threshold for each pixel by calculating the weighted average of the neighborhood of the pixel [26], and uses the local threshold to process the current pixel, which can be expressed by

$$t(x, y) = a \bullet \mu(x, y) + b \bullet \sigma(x, y) \tag{7}$$

where $a$ and $b$ are non-negative constants, $\mu(x,y)$ and $\sigma(x,y)$ are the mean value and the standard deviation value of neighborhood of the pixel $(x,y)$ respectively.

This paper uses OpenCV function $cv::adaptiveThreshold$ and the calculation method of local threshold $t(x,y)$ can be abstracted as

$$t(x, y) = F(x, y) - c \tag{8}$$

where $F(x,y)$ represents the convolution value at the pixel $(x,y)$ using the filter kernel, $c$ is a constant value, which can be determined by experience or debugging the system.

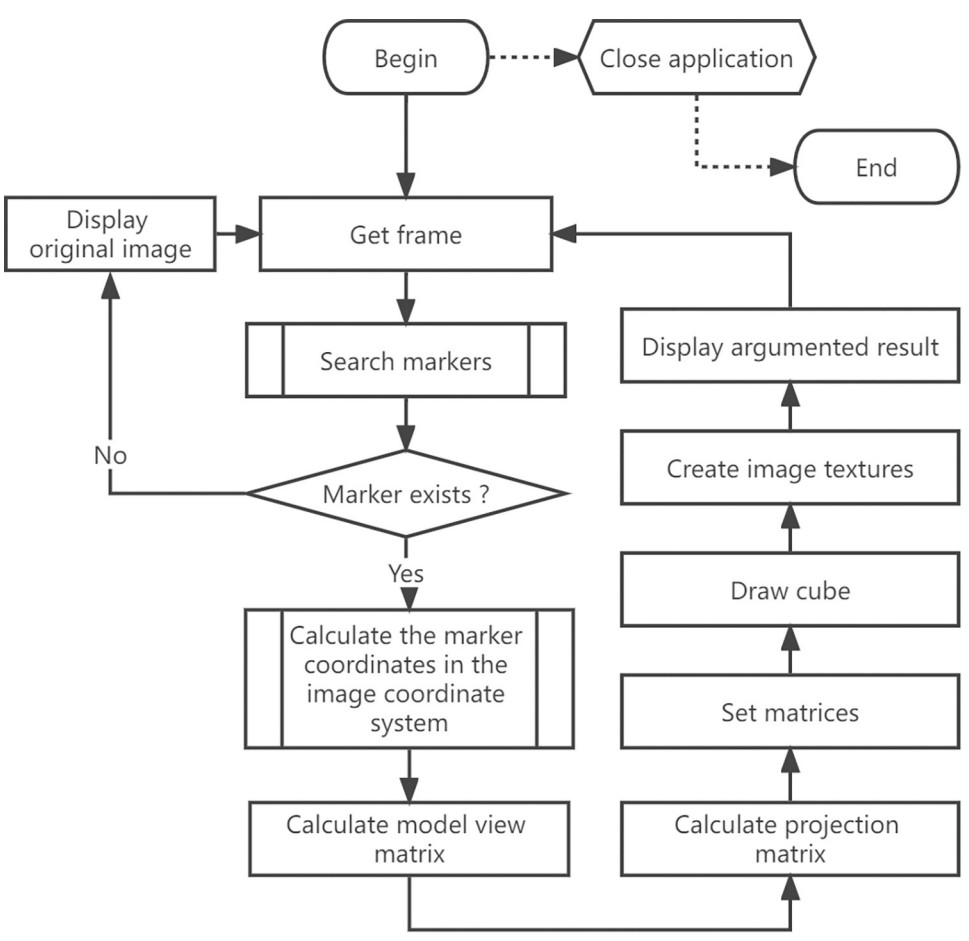

**Fig 3. Major processing flow.**

In this paper, Gaussian filter kernel is used. At the same time, the operation of inverse binarization is used in view of the characteristics of low overall pixel value and high surrounding pixel value of the marker, as a result, the pixels greater than the threshold are set as background pixels and the others are set as foreground pixels. The binarization result is shown as Fig 5(B).

**Step 3** Remove image noises and extract features. Here, the open operation of morphology is adopted, that is, the image is processed by the erosion operation followed by a dilation operation. The result is shown as Fig 5(C).

**Step 4** Detecting corners, whose results are a series of point groups. This paper uses the OpenCV library function *cv::findContours* to find corner groups. For each point group, judge whether it meets the basic conditions of marker's posture, such as whether it is a convex quadrilateral, whether the length of the four edges is within a reasonable interval, and so on. If the basic conditions are not met, exclude the point group, otherwise continue to the step 5.

**Step 5** Obtain the image perspective transformation matrix. For the point group that meets the basic conditions of predefined marker, the perspective transformation matrix of the marker image is obtained by using the OpenCV library function *cv::getPerspectiveTransform* with the coordinate matrices $M_c$ and $M_w$.

**Step 6** Use the OpenCV library function *cv::warpPerspective* to perspective transform the gray image and extract the image area where the marker is located (hereinafter referred to as marker image). The extracting side length equals to $L_M$.

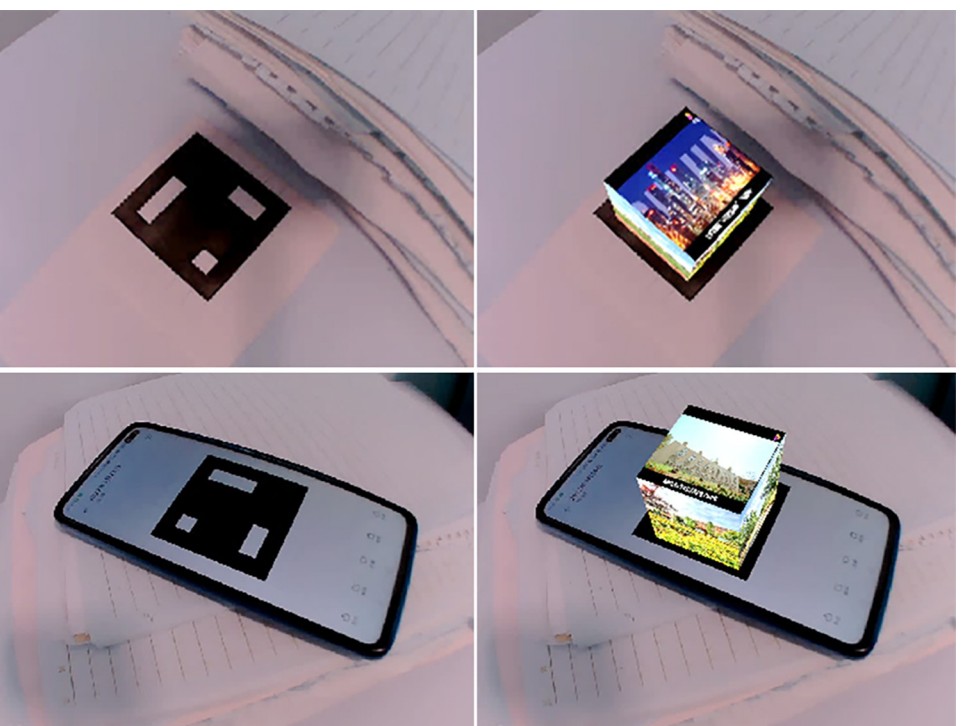

**Fig 4. Original image and result after being augmented.**

**Step 7** Binarization of marker image. It can be seen from Fig 5(A) that affected by the influence of camera parameters, ambient light and other factors, the pixels of the marker image are not the two extreme values of black and white. Therefore, binarize the image after it is obtained. Different from the image in step 2, the processed object at this time is the marker image excluding other areas. The marker image should be binarized as a whole because its histogram has two typical peaks [27].

Set the proportion of foreground pixels of the image as $\omega_f$, the average gray value as $\mu$, the proportion of background pixels is $\omega_b$ and the average gray value is $\mu_0$. The four parameters all are the function of the segmentation threshold $t$ of foreground and background. Then the variance between the total gray levels can be expressed as

$$\sigma^2(t) = \omega_f \bullet \omega_b(\mu_f - \mu_b)^2 \tag{9}$$

The pixel proportion of foreground and background and its average gray level will change when the threshold $t$ changes, which will affect the value of inter class variance finally. When the inter class variance reaches the maximum, the segmentation threshold $t$ is the best threshold. This is the maximum interclass variance algorithm, also known as OTSU algorithm. It can adaptively determine the optimal threshold of binarization according to the information of the image. The corresponding function *cv::threshold* of OpenCV library provides the input argument option of OTSU algorithm. The gray marker image and its binary image are shown in Fig 5(D) and 5(E).

**Step 8** Identify the marker content. Even if the optimal binarization threshold is found through the adaptive method in the step 7, not all pixels of each cell are as expected after the binarization, that is, there are black pixels in white cells and white pixels in black cells, as shown in Fig 5(E). Consequently, count the number of non-zero pixels in each cell of the

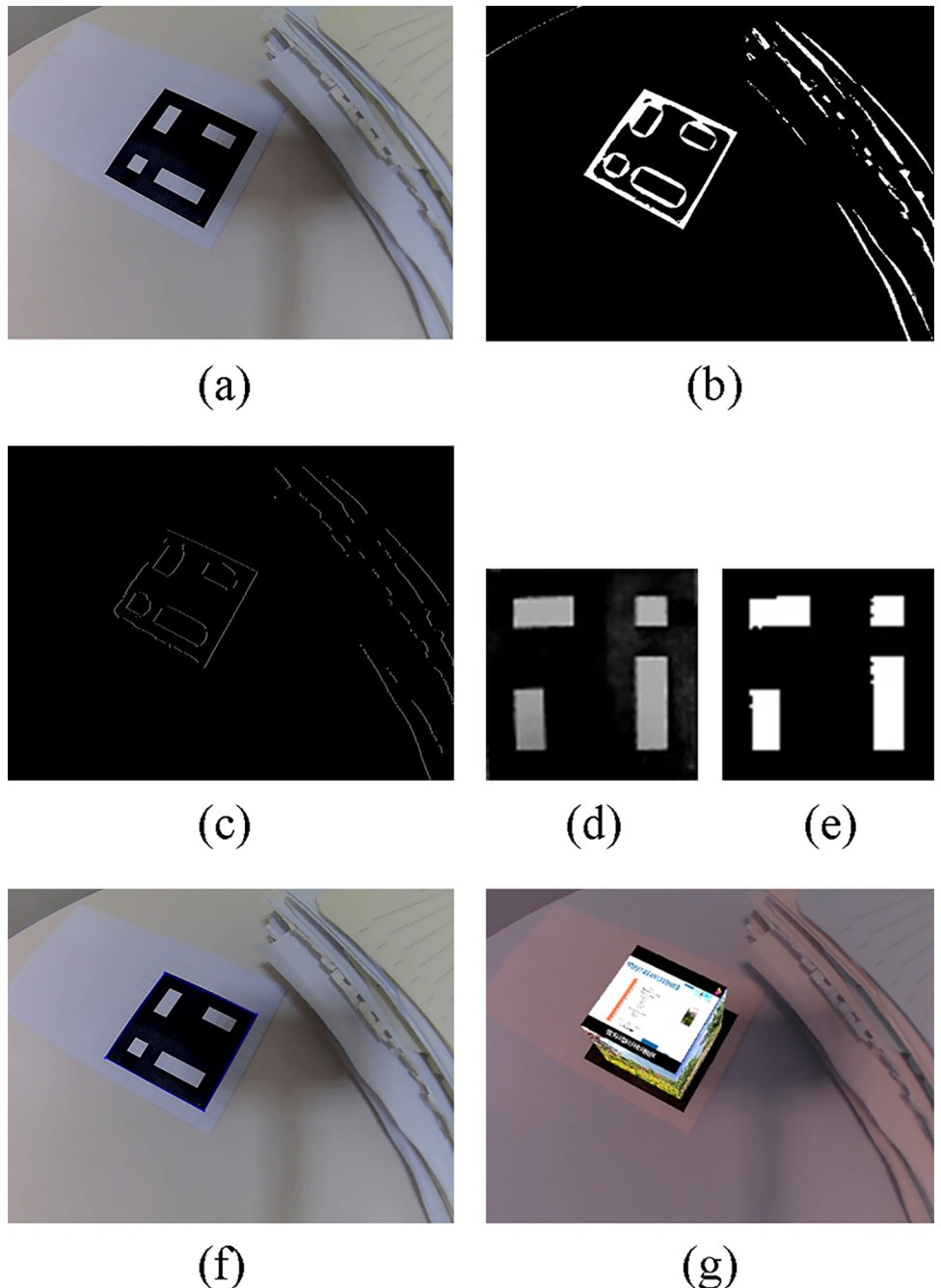

**Fig 5. Marker recognition.** (a) Original image. (b) Binarization image. (c) Feature extraction. (d) Marker. (e) Binary image of marker. (f) Corner location. (g) Augmented results.

marker $N_r$ by means of OpenCV function $cv::countNonZero$, and set a certain threshold $N_t$ from the count of all pixels in the cell $N_w$, such as three quarters of $N_w$, to confirm whether the color of each cell is white or black. The color of the cell is white if

$$N_t > N_r \qquad (10)$$

or black if

$$N_t \leq N_w - N_r \tag{11}$$

Then the information of each cell (white means 1, black means 0) and the information matrix of the whole marker are obtained. Continue to the step 9 if the definition of the marker is met, otherwise, exclude the corner group.

**Step 9** Set the isolated cell as the upper left corner cell according to the definition of the marker to calculate the plane rotation state of the current marker. The information matrix of the marker is also updated while solving the plane rotation of the marker.

**Step 10** Now, the coordinates and order of the four corner points of marker are obtained, and then the corner points can achieve sub-pixel accuracy to obtain more exact coordinates. We make an image of Fig 5(E) to show the coordinates of the corner on the original image. The black frame is displayed together with the marker for ease of elucidation.

The algorithm flow of marker recognition is shown as Figs 5(G) and 6 shows the final augmented effect, from which we can see that the virtual cube is accurately created on the marker.

## Texture creation

The system can display both images and videos data. The data can be existing files or collected in real time through cameras. The application obtains the displaying images or videos data through the configuration file. For the display of images, the image texture can be set on the faces of the virtual cube.

In order to display the existing advertising videos on the faces of the virtual cube or display the video stream of the current camera in real time (customers can see themselves by this way), the frame data of the video stream is obtained when the OpenGL window data is updated. The image textures are created from the images by using the OpenGL library function and set on the faces of the virtual cube. Through the continuous updating of frames, the textures displayed on the cube faces will also change, so as to achieve the purpose of playing videos on the faces of virtual cube.

Since the image textures are created constantly from the frames of the videos, they should be deleted by the application after use, otherwise it will continue to occupy the memory and have a negative impact on the operation of the system.

## Interface design

The interactive window is used to display the final result. If there are predefined markers appearing in the frame collected by the camera, the interactive window will display the original image and the virtual object. Otherwise, only the original frame of the world collected by the camera is displayed on the background. The implementation is to create a graph window inherited from *QGLWidget* and rewrite its *initializeGL*, *paintGL* and other useful functions. The initialization data such as camera parameters and standard marker definition is set in the *initializeGL* function; the real-time marker detection, virtual objects creation, and display of the combination of virtual and real data are set in the *paintGL* function.

## Interactive gesture operation

### Motivation

The system can display different advertising videos on the six faces of the virtual cube. It can be predicted that the videos on some faces will be blocked by other faces. Moreover, in most cases, the video display area is only a small part of the entire screen area, and the advertising

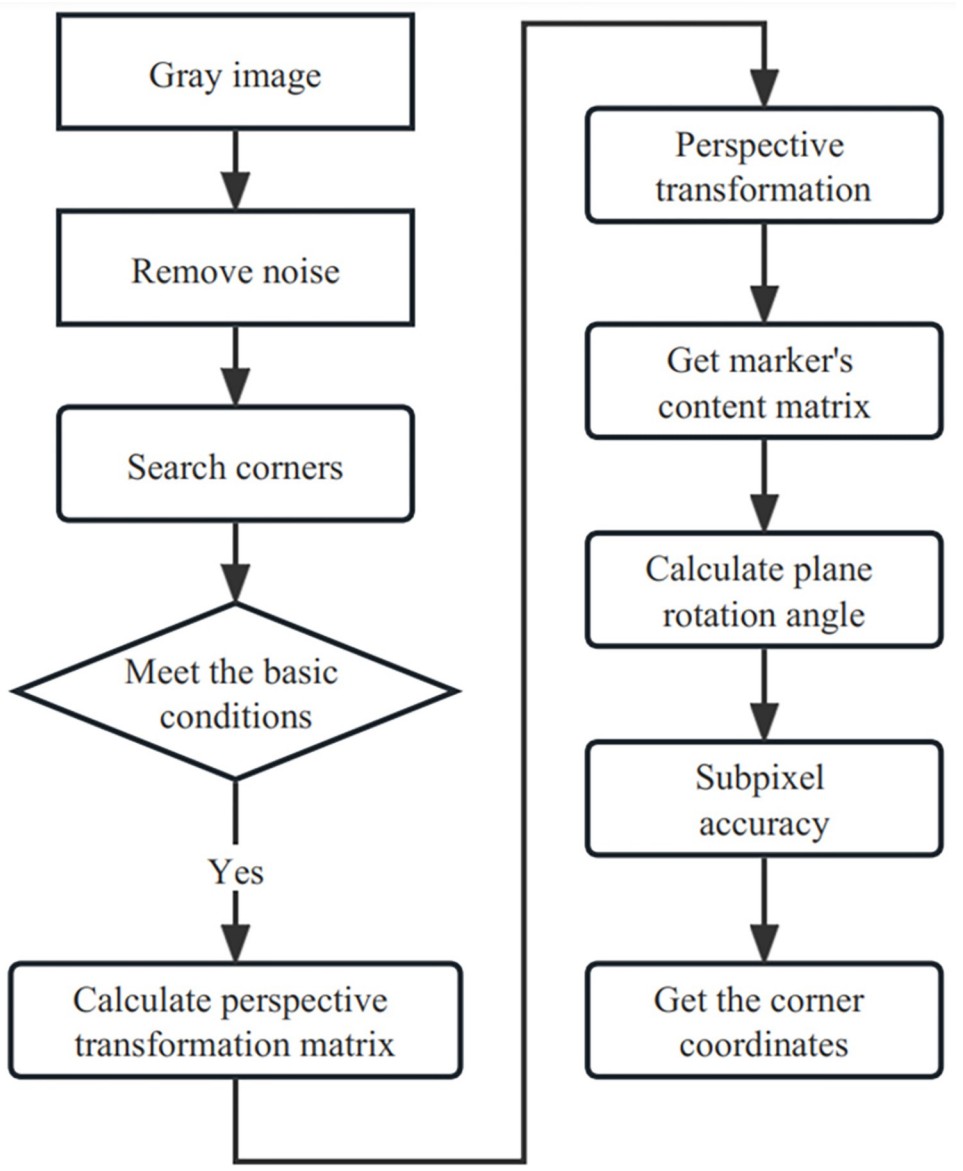

**Fig 6. Marker recognition algorithm flow.**

content may not be clearly displayed because the playback area is too small. Therefore, the system supports the gesture operation of the virtual cube to move, rotate and zoom in or out the virtual cube. The virtual cube always follows the marker if the marker exists in the view of the camera, otherwise, it will maintain its final posture and wait for the intervention of gesture instruction.

There are many mature frameworks for gesture recognition [28–32], but most gestures cannot be used directly by this system. We select MediaPipe Hands [31] to extract the landmarks of hands, and on this basis, to design a set of gestures suitable for this system. MediaPipe Hands was open source by Google research in 2019. It supports five fingers and gesture tracking, and can infer the 3D-coordinates of 21 landmarks of the hand as shown in Fig 7 from a frame. It has high robustness even if the partial display of the palm or the hands self-occluded, and the comprehensive recognition accuracy has reached 95.7%, which has the characteristics

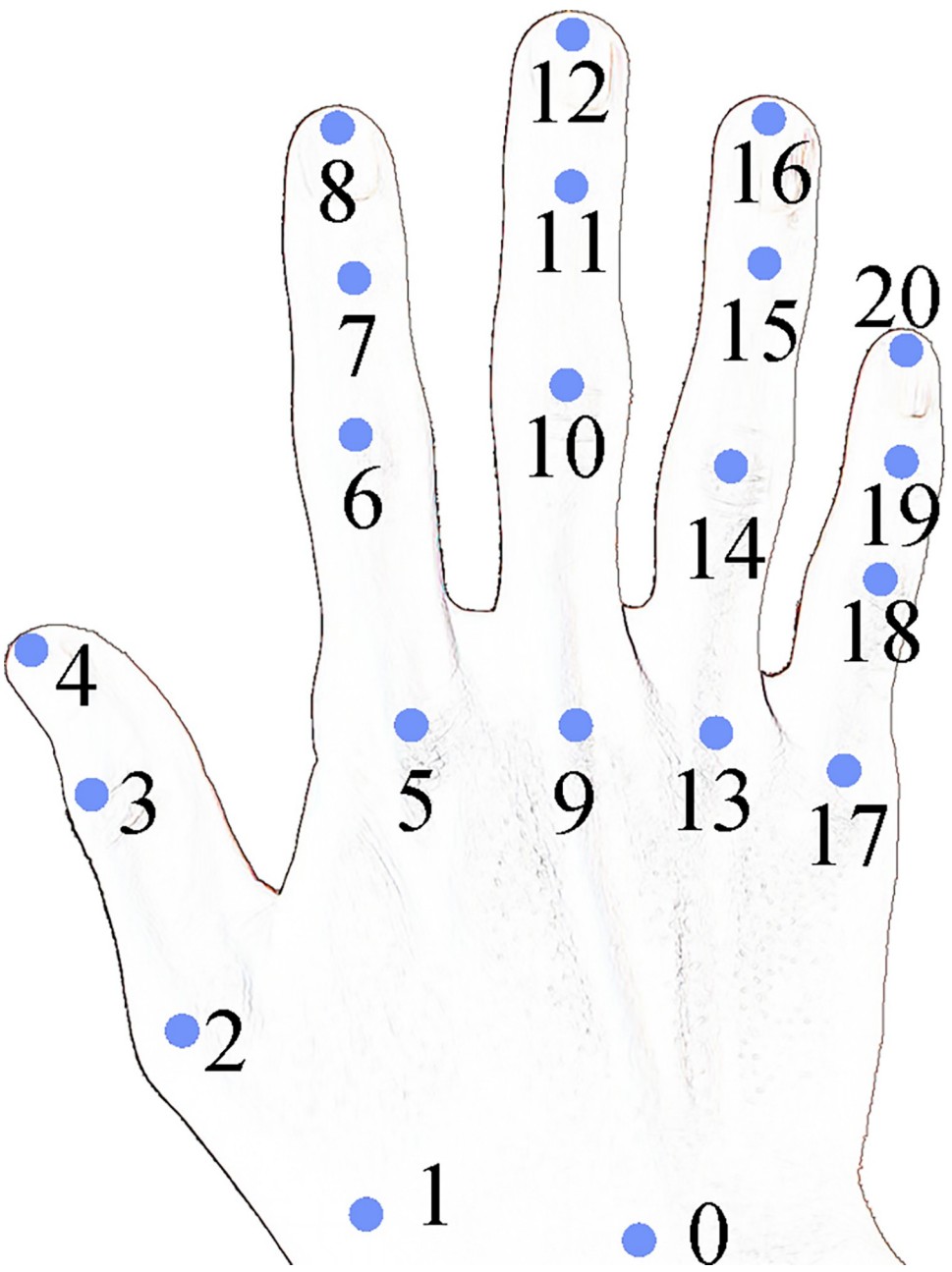

**Fig 7. The 21 landmarks of a hand.**

of high performance and low time consumption. Based on this, it has been applied in many research [33–37].

## Mapping of operation area and screen area

As shown in Fig 8, the range of the screen is $M_0$ and the active area of the index fingertip is $M_1$. The hand recognition rate will decrease if the index finger tip is out of $M_1$, due to incomplete display of hands in a single frame. It is necessary to map the fingertip coordinates in $M_1$ to $M_0$ to calculate the coordinates of the hand relative to the whole screen. The X coordinate of the

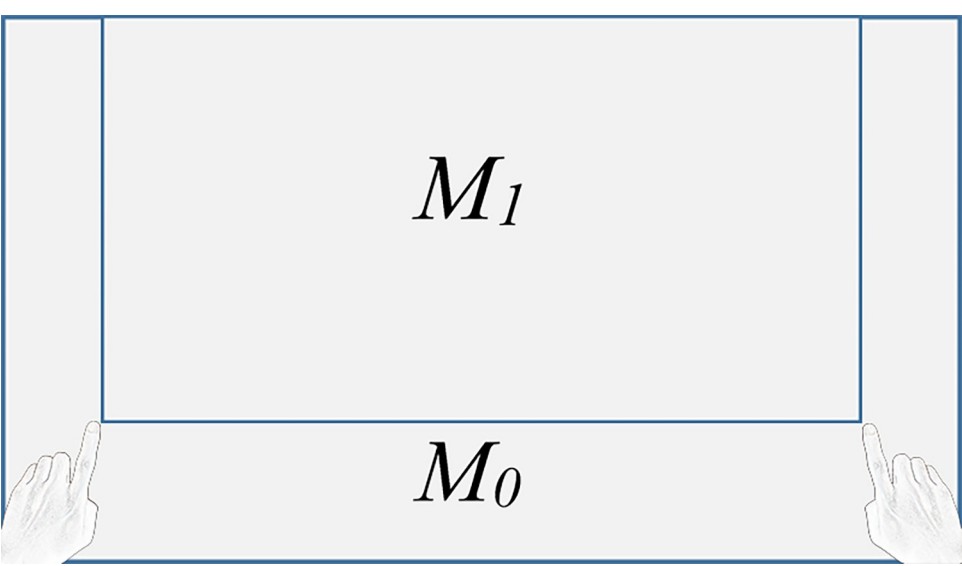

**Fig 8. Operation area and screen area.**

mouse pointer meets the following formula

$$x = \begin{cases} 0 & x_c \in [0, x_r] \\ W & x_c \in [x_r + w, W] \\ W/w \bullet (x_c - x_r) & x_c \in (x_r, x_r + w) \end{cases} \quad (12)$$

where $W$ is the screen pixel width, $w$ is the width of the fingertip active effective area, $x_c$ is the current fingertip coordinate, and $x_r$ is the X coordinate of the starting point of the fingertip area.

When the fingertip is on the left side of the effective area, the X coordinate of the mouse pointer is 0, indicating that the pointer is considered to be on the leftmost side of the screen. Similarly, when the fingertip is on the right side of the effective area, the X coordinate of the mouse pointer is the width of the screen resolution. When the fingertip is inside the effective area, the X coordinate of the mouse pointer is the linear mapping value.

For the same reason, the Y coordinate of the mouse pointer meets

$$y = \begin{cases} 0 & y_c \in [0, y_r] \\ H & y_c \in [y_r + h, H] \\ H/h \bullet (y_c - y_r) & y_c \in (y_r, y_r + h) \end{cases} \quad (13)$$

where $H$ is the height of the screen pixel, $h$ is the height of the active area of the fingertip, $y_c$ is the current fingertip coordinate, and $y_r$ is the Y coordinate of the starting point of the fingertip area. In Fig 8, $y_r = 0$ since $M_1$ on the top of $M_0$.

In fact, the setting of area $M_1$ needs to consider many factors. The swing amplitude of the user's arms will be increased if the area is too large, which may increase the user's fatigue. The area cannot too small because of the mapping from the finger active area to the whole screen area, otherwise, small changes in the effective area $M_1$ will have great feedback on the whole screen $M_0$, which will affect the positioning accuracy. We have tested and verified that it will be better when the area of the operation area $M_1$ is about half of the screen area $M_0$.

## Mouse sensitivity design

There are many sample or mature applications at https://www.github.com, but most applications use the absolute position of one fingertip as the mouse pointer, which has two disadvantages. Firstly, because the human hands cannot remain absolutely stationary in front of the camera, when operating with gestures, the small shaking of the hand may lead to the irregular transformation of the virtual cube. At this time, it is necessary to eliminate the possible hand shaking error. If the distance moved is within a certain error range, it is considered that the movement is an invalid signal caused by the shaking of the hand. This method generally has a filtering effect with the shaking of the hand, but when the user really wants to make a small adjustment within the error range, the signal will be ignored as an error. Secondly, for the sake of the filtering hand shaking, when using this algorithm for gesture operation, the movement of the mouse pointer is not continuous but jumping. The step size of jumping is related to the set error value, which will bring users the feeling of insufficient fluency.

For the coordinates setting of the mouse pointer, this system detects the coordinates of the index fingertip on the screen, and takes the fingertip coordinates and the last mouse pointer position as the weighted value as the new pointer position.

The weighted results of fingertip coordinates $P(x_r, y_r)$ may have values beyond the screen range, so the following constraints need to be imposed on the weighted results for getting the final mouse pointer position

$$x_r = \begin{cases} 0 & x_r < 0 \\ W & x_r > W \\ x_r & 0 \leq x_r \leq W \end{cases} \tag{14}$$

$$y_r = \begin{cases} 0 & y_r < 0 \\ H & y_r > H \\ y_r & 0 \leq y_r \leq H \end{cases} \tag{15}$$

## Gesture unit definition

We defined two status of each finger, extended one and closed one. The status represents a signal of each finger and every finger has its independent signal. The final gesture is the result of all five finger signals. The gesture of a single finger is called a gesture unit. For example, the V-shaped gesture is formed by combining the index finger and the middle finger at an angle.

For the thumb, we estimate the direction of the hand first, namely, estimating the front or back of the palm facing the camera. Then, give the following definition for the determination of finger extension

$$X(P_4) > X(P_2) \tag{16}$$

if the front of the palm facing the camera, otherwise,

$$X(P_4) < X(P_2) \tag{17}$$

where $X(P_2)$ and $X(P_4)$ are landmarks of the thumb as shown in Fig 7.

For the other four fingers except the thumb, we give the following definition for the determination of finger extension

$$Y(P_i) < Y(P_{i-2}) \tag{18}$$

where $i \in [8,12,16,20]$ is the index of the fingertip landmarks as shown in Fig 7. $P_i$ is the four fingertips.

For example, the condition for judging the extension of the index finger means that the Y coordinate value of the 8th point is less than the Y coordinate value of the 6th landmark, namely $Y(P_8) < Y(P_6)$.

This estimation method allows users to use a variety of fine-tuning gestures to express the same signal during operations, which can reduce the fatigue when making gestures continuously.

### Gesture operation design

**Operation status.** The system defines several states, such as initial, front view, translation, rotation, zooming and pause. Gestures represent different meanings in different states even if they are the same. The gestures of the clenching fist as shown in Fig 9(A) and the all five fingers extension as shown in Fig 9(B) make up the signal of state switching. The system switches the states when detecting the gesture of all five fingers extension with the condition that the last gesture is the gesture of the clenching fist.

**Fronting operation.** Fronting refers to the operation of placing one face of the cube parallel to the screen, so that the advertising content on one face can be displayed alone, while the other faces are all blocked and not displayed. Fronting operation does not need to recognize the change of gestures, but only the current static ones. In this paper, the six gestures as shown in Fig 9(B)–9(G), are used as the signals of the fronting operation on the six faces.

**Translation operation.** Different from the fronting operation, the translation operations need a magnitude to calculate the measure of the transformation. This magnitude is generally the position variation of the index fingertip, which is regard as a mouse pointer. The gesture operation on virtual objects is to simulate the operation of the physical mouse on it.

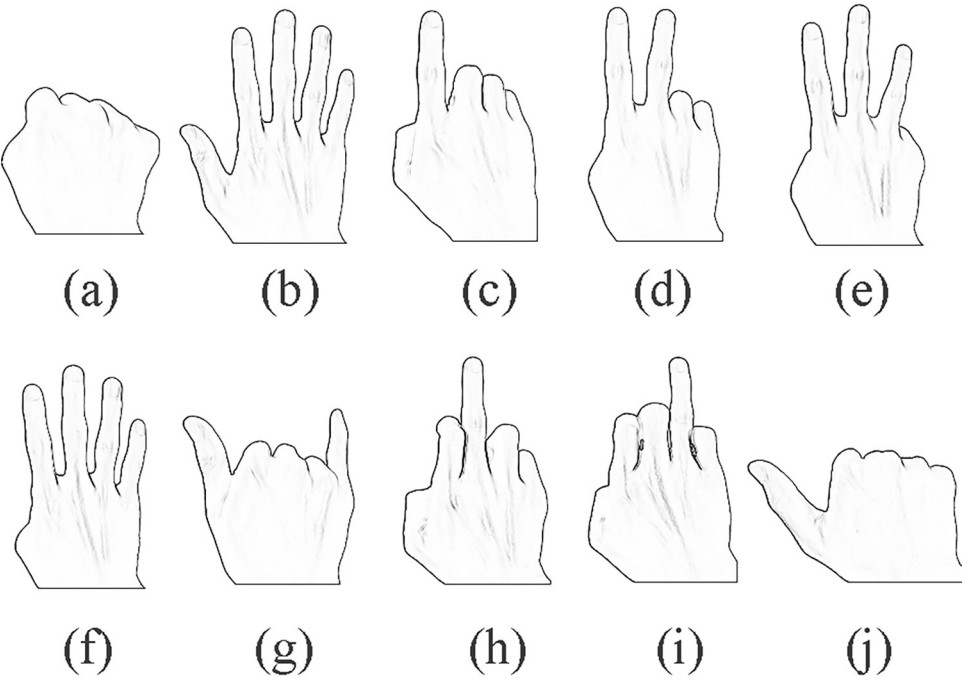

**Fig 9. Gestures are used.**

**Rotation operation.** When in the rotation state, the index finger as shown in Fig 9(C) expresses the signal of the rotation around the X axis, the middle finger as shown in Fig 9(D) expresses the signal of the rotation around the Y axis and the fourth finger as shown in Fig 9 (E) expresses the signal of the rotation around the Z axis. The axes are independent and can be combined to rotate the virtual cube around any axis. If the thumb as shown in Fig 9(J) is extended at the same time, it reverse rotation around the axes.

**Zooming operation.** In the zooming status, virtual cube was zoomed in under the signal of index finger as shown in Fig 9(C). Similarly with rotation operation, if the thumb as shown in Fig 9(J) is extended at the same time, virtual cube was zoomed out.

**Operation tips.** In order to make customs who first contact with the system more clearly know the current state of the system and the operations that can be carried out under the current state, the system provides the virtual state buttons and operation text prompts on the human-computer interface. All states have a unique translucent button, which can show users simple and effective operation tips without blocking the main window. The button transparency of the currently active state is low to highlight the state. Below the virtual button, the operations that can be performed in the current state are displayed in order by line, and the ongoing operations are highlighted. The operation prompts can effectively reduce the error rate of operators, especially those who first contact the system, and increase the friendliness of the system.

## System evaluation and test

### System performance

The marker designed in this paper is a two-dimensional code and the colors are only black and white. The ambient light has little influence on it. Therefore, the accuracy of the marker recognition has reached 95% and most failure scenarios are due to the poor angle between the camera and the marker. It can be seen from the above system flow that when identifying the marker corners, the time complexity is related to the number of corner groups in the frame. Therefore, the theoretical time complexity of the marker recognition is $O(n)$. However, due to the relatively perfect pre-processing before marker corner recognition, the count of corner groups are less than 5 in most cases in fact, so the algorithm complexity is close to $O(1)$. On the other hand, the pipeline rendering also has a complexity $O(1)$ because the efficiency and the algorithm complexity of the pipeline rendering depend on the count of videos only, which has the max value six. Therefore, the whole time complexity of AR box creating is close to $O(1)$.

For the real-time interactive augmented reality system, the accuracy and execution efficiency of marker recognition are the most important factors affecting the customers' experience [38, 39]. The final output videos are inconsistent if the application is of a poor efficiency, for this reason the customers' experience is poor. Similarly, the combination of virtuality and reality cannot be achieved if the recognition accuracy is low, which makes it worse of the customers' experience. For the efficiency, different gestures have the same time on extracting landmarks of hands by MediaPipe Hands, which spends 8ms(CPU: Intel Core i7-7700HQ, graphics card: Intel HD graphics 630), including the time of reading frame. For the accuracy, the system has different values on recognizing different gestures, as shown in Fig 10, with the maximum value 100%, minimum value 95% and average value 97.7%. The comprehensive accuracy may be higher because the gestures with higher recognition accuracy are also used more frequently. The dataset that supports the result is available at https://github.com/wanzhuxie/ARIAS/tree/PLOS-ONE/minimal data set.

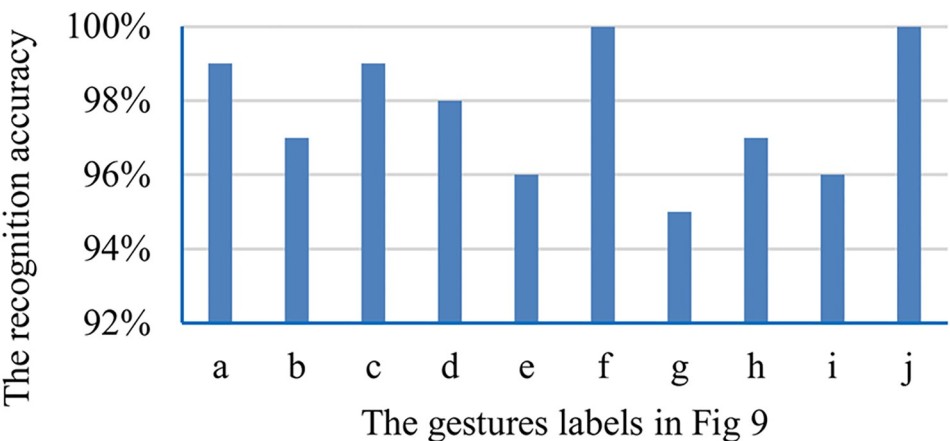

**Fig 10. Gesture recognition accuracy.** a-j correspond to the gestures in Fig 9.

## Methods

**Ethics statement.**   The study has been approved by the Foundation Department of Southwest Jiaotong University Hope College and it have been conducted according to the principles expressed in the Declaration of Helsinki for experiments involving humans. All of the participants gave a written informed consent indicating that they took part in the study voluntarily. The individual in this manuscript has given written informed consent (as outlined in PLOS consent form) to publish these case details.

**Test and results.**   Most consumers may not willing to spend too much time on the learning of the advertising system. How to make the consumers learn the way to interact with the system when they first contact the system becomes particularly important. In order to get the ability of the system to be accepted by users, we invited 22 people to do the system experience test.

The participants firstly show card marker or electronic marker in front of the camera, then the virtual cube appear and can be manipulated by gestures. After the virtual cube appear, the participants make corresponding gestures and manipulate the virtual cube at the request of the tester, as shown in Fig 11. Each subject carried out six rounds of tests. The system has a total of 26 operation instructions. It can be expected that when the system is really used, the users generally only use a few of them. And when the users are operating, the actual operation sequence is not consistent with that in the system interface, but is related to the posture of the virtual object. Therefore, in order to obtain the data of the user's actual operation, only 10 of the operation instructions are randomly selected for testing during each round.

We use two indicators to evaluate the learnability of the system. One is the number of errors in each round of testing, and the other is the time it takes to complete all the required instructions in each round. Gesture errors include not only making wrong gestures, but also the recognition errors caused by correct gestures but inappropriate angles relative to the webcam.

The Table 1 shows the number of errors of each participants in each round of test and the Fig 12 shows the curve corresponding to these error number data. There are 22 curves in the Fig 12, but only some of them were seen because some segments of different curves are overlapped. The thicker the segments, the more curves overlap here. The Fig 13 shows the average of error number of all the participants in each round of test.

Similarly, The Table 2 shows the time taken of each participants in each round of test and the Fig 14 shows the 22 curves corresponding to these time taken data. The Fig 15 shows the average time taken of all the participants in each round of test.

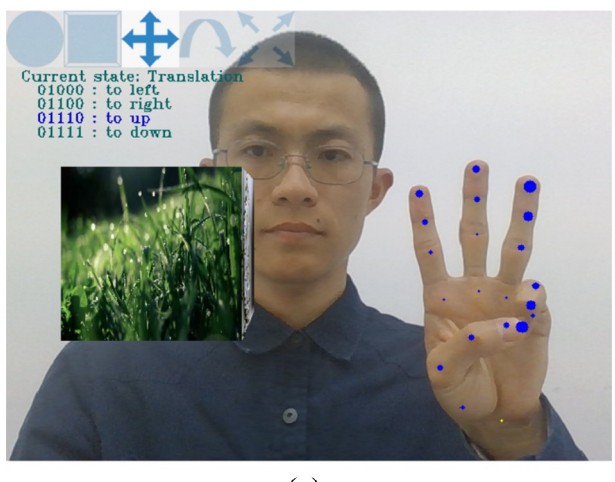

**Fig 11. The system testing scenes.** (a) showed the moving up in the translation state. (b) showed the rotating around the X and Y axes simultaneously in the rotation state. The hand landmarks have different sizes, which represent the Z coordinate value with the wrist landmark as the zero point. The larger the diameter of the circle, the greater the absolute value of the distance. Blue means closer to the camera, and yellow means farther.

## Potential applicability

In order to obtain more specific potential adaptability of the system, after the participants learned and operated the system, we invited them to do a questionnaire survey. There are 10 objective selection questions, as shown in the Table 3. Among them, questions 1–6 relate to the potential effect of the system compared with traditional advertising forms, and questions 7–10 relate to the learnable of the system. There are 220 statistical results, and the number of YES, NO and N/A (neither or no answer) are 189, 14 and 17 respectively.

## Discussion

### Analysis of test results

We expect that the number of errors and time taken will gradually decrease with the increase of test times before the testing. In fact, quite a few individual test data are not as we expected. No matter the number of errors or the time taken to complete the instructions, they do not

**Table 1. The number of errors of each participants.**

| User ID | 1st | 2nd | 3rd | 4th | 5th | 6th |
|---------|-----|-----|-----|-----|-----|-----|
| user1 | 0 | 0 | 0 | 0 | 0 | 0 |
| user2 | 0 | 0 | 0 | 1 | 0 | 0 |
| user3 | 1 | 1 | 1 | 0 | 2 | 1 |
| user4 | 1 | 2 | 0 | 0 | 1 | 0 |
| user5 | 0 | 1 | 1 | 1 | 1 | 1 |
| user6 | 0 | 0 | 1 | 0 | 0 | 0 |
| user7 | 0 | 0 | 0 | 0 | 0 | 1 |
| user8 | 3 | 3 | 2 | 1 | 2 | 1 |
| user9 | 0 | 0 | 1 | 1 | 0 | 0 |
| user10 | 1 | 1 | 0 | 0 | 0 | 0 |
| user11 | 3 | 3 | 2 | 2 | 0 | 1 |
| user12 | 3 | 1 | 1 | 0 | 0 | 0 |
| user13 | 0 | 0 | 1 | 0 | 0 | 0 |
| user14 | 1 | 2 | 0 | 1 | 0 | 1 |
| user15 | 3 | 5 | 3 | 3 | 4 | 4 |
| user16 | 2 | 0 | 1 | 2 | 1 | 0 |
| user17 | 2 | 1 | 2 | 1 | 1 | 0 |
| user18 | 1 | 0 | 0 | 0 | 1 | 0 |
| user19 | 1 | 1 | 0 | 1 | 0 | 1 |
| user20 | 0 | 2 | 0 | 1 | 2 | 2 |
| user21 | 2 | 0 | 1 | 1 | 0 | 1 |
| user22 | 1 | 0 | 1 | 0 | 0 | 0 |

gradually decline with the number of tests, but have obvious fluctuations. For the fluctuation of the number of errors, we think it is normal. Most of the fluctuation values are only 1, that is, one test has only one error, compared to the previous one. On the other hand, the system is not complicated, and we also provide virtual buttons and text prompts to remind the user of the current status and available operations. The number of errors of most participants is not high, only once or twice, so the fluctuation of the error number is unavoidable. It can be assumed that if the base number of errors is ten or even dozens of times, the single curve may has no fluctuation or it is likely to fluctuate slightly. For the fluctuation of operation time, we think it is related to random instructions, specifically, there are two possible reasons. First, no state switching is required among the continuous operations in the same status, when the next command is operated, it only needs to make the corresponding gesture immediately, and thus the operation time is less. However, when continuous operations in different states, it needs to switch states before making the next gesture. Sometimes, it even needs to switch to multiple states, which consumes more time. Second, some gestures are easy to make, such as only extending the index finger, while others are not easy to make, such as those involving the ring finger. Customers make gesture operations according to the posture of the virtual object, which changing constantly with the posture of the marker, these gestures thus cannot be predicted in advance. Therefore, we believe that the fluctuation of the test results also corresponds to the real one.

However, the average value of the errors and the time taken both have a downward trend. The error number curve has a steep decline between 1–3 times, whereas the time taken curve has an obvious steep decline between 1–2 times and then tends to be stable. This may be able to explain that the learnability of the system is good, while most users can basically master the

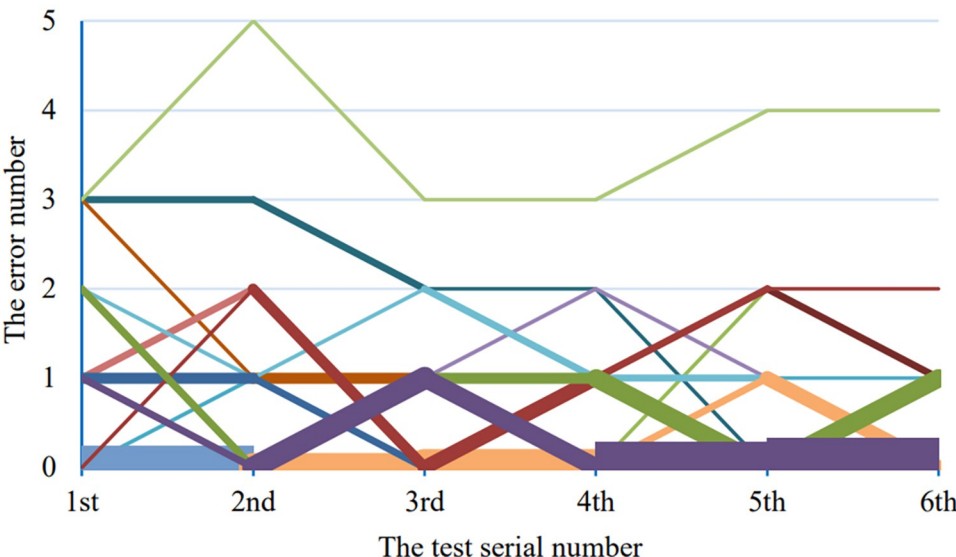

**Fig 12. The error curves of each participants.** The horizontal axis indicates the test serial number and the vertical axis indicates the number of errors.

operation of the system with 1–2 attempts. At the same time, from Figs 12 and 14, we can also see the differences in the adaptability of the system among individuals. Some users can get started quickly, while others are not.

The vast majority of the participants showed a strong interest in the system. Compared to traditional advertising forms, Table 3 shows that the positive responses accounted for 93.1% of the total positive and negative responses, indicating that the system is highly popular and has high potential applicability and promotion. At the same time, most of the participants showed great expectations for the system and put forward many useful suggestions. There are quite a few participants believe that the attractiveness of the system needs to be strengthened, and competitive game factors can be added to attract customers; half of the participants think that the gestures related to the ring finger are difficult to make, which should be improved.

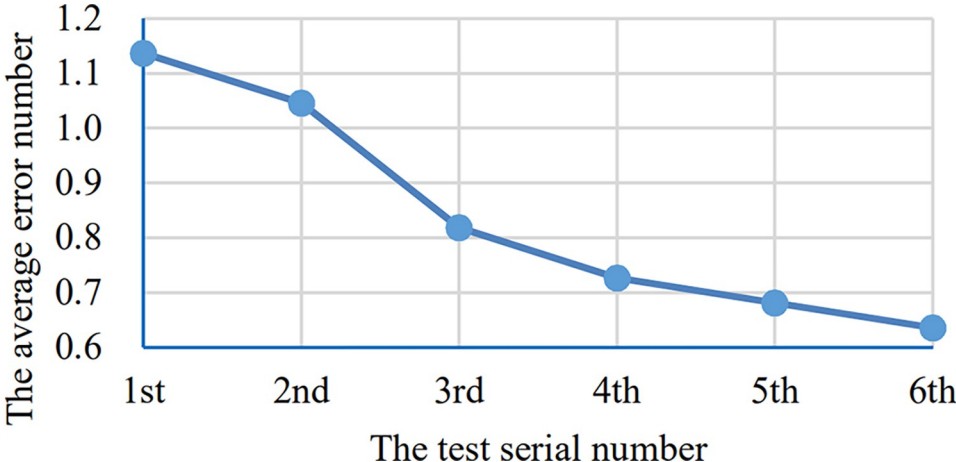

**Fig 13. The average error curve.** The horizontal axis indicates the test serial number and the vertical axis indicates the average error number of all the 22 participants.

**Table 2. The time taken data.**

| User ID | 1st | 2nd | 3rd | 4th | 5th | 6th |
|---------|-----|-----|-----|-----|-----|-----|
| user1 | 39 | 33 | 46 | 36 | 33 | 44 |
| user2 | 44 | 31 | 29 | 43 | 38 | 47 |
| user3 | 41 | 53 | 43 | 39 | 48 | 45 |
| user4 | 50 | 50 | 51 | 54 | 58 | 57 |
| user5 | 78 | 57 | 53 | 62 | 66 | 54 |
| user6 | 45 | 44 | 54 | 36 | 52 | 40 |
| user7 | 46 | 46 | 43 | 35 | 35 | 40 |
| user8 | 76 | 63 | 62 | 67 | 93 | 66 |
| user9 | 57 | 47 | 58 | 59 | 42 | 37 |
| user10 | 66 | 52 | 61 | 50 | 45 | 47 |
| user11 | 103 | 81 | 90 | 68 | 65 | 62 |
| user12 | 85 | 65 | 63 | 64 | 56 | 53 |
| user13 | 65 | 56 | 50 | 51 | 47 | 48 |
| user14 | 106 | 84 | 65 | 71 | 53 | 65 |
| user15 | 62 | 72 | 79 | 62 | 62 | 74 |
| user16 | 86 | 75 | 58 | 81 | 50 | 59 |
| user17 | 65 | 43 | 51 | 58 | 64 | 62 |
| user18 | 70 | 47 | 44 | 48 | 43 | 39 |
| user19 | 57 | 48 | 53 | 47 | 52 | 56 |
| user20 | 46 | 58 | 55 | 48 | 50 | 46 |
| user21 | 80 | 44 | 50 | 45 | 33 | 41 |
| user22 | 69 | 58 | 52 | 55 | 62 | 63 |

## System features

The effect of this system belongs to the category of augmented reality, which has obvious advantages and disadvantages compared to the mixed reality technology. The technical principle of mixed reality is similar to that of augmented reality, they both overlay virtual objects on physical objects. The difference is that they present the results in different ways. The mixed reality system generally uses the headset, such as Microsoft HoloLens [40], as the necessary auxiliary equipment, which has both cameras and monitors, the glasses are the monitors in fact. The operator can directly see the virtual objects overlaid on the real world, rather than the mirror effect displayed through the large monitor, which makes the operator more naturally integrate into the scene of virtual and real combination. Moreover, some headsets can recognize the hand gestures and/or gaze data of the operator. For gesture recognition, the recognition effect of the headset is basically consistent with this system in the theoretical domain, with the principle that to extract and analyse gestures after collecting hand images. For the recognition of gaze data, there is one or more cameras (generally two) dedicated to collecting the eyes data. In theory, the accuracy of gaze recognition by the headset will be higher than that of this system because the positions of eyes relative to the cameras are always fixed, no matter how to shake the head and change the gaze directions of eyes. Therefore, the eye positions on the images captured by the headset remain unchanged, which is easier to process the collected image.

While the headset is effective in facilitating operators' natural integration in the mixed reality scenario, its cost and public health need to be considered. When multiple customers must be taken care of simultaneously, multiple sets of equipment are required, and each customer may need to be explained by a professional. In terms of the public health, it remains to be

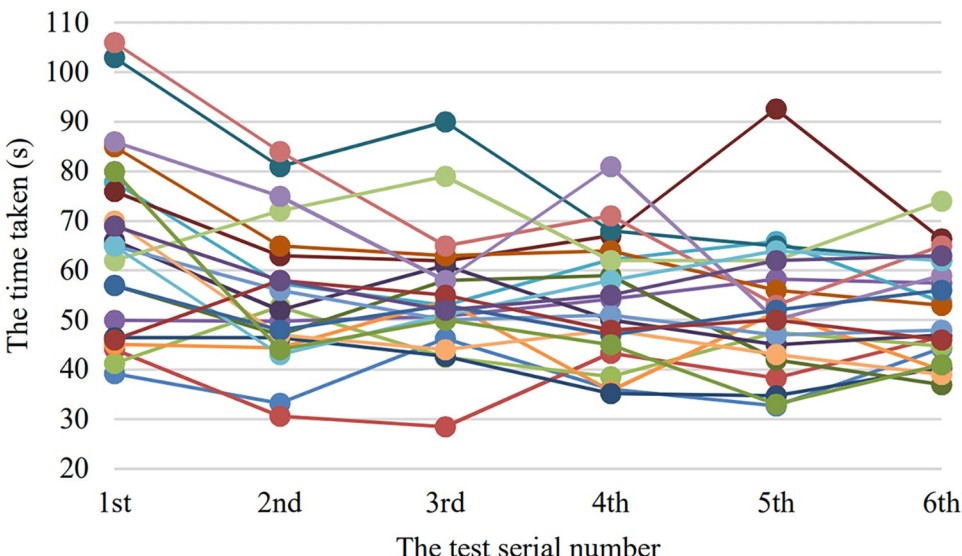

**Fig 14. The time taken curves of each participants.** The horizontal axis indicates the test serial number and the vertical axis indicates the time taken in seconds.

considered that whether customers are willing to wear the headset that has been worn by many others, especially when COVID-19 has the possibility of epidemic today. This system requires only one camera to collect gesture data, and the result of the virtual and real combination is displayed on a normal monitor with a lower hardware cost. Even customers who are not in operation can see the effect of combining virtual with real, and one professional can explain the use of the system to multiple customers at the same time. Moreover, all operations of the system can be completely contactless and customers are free from health issues. Therefore, this system is more suitable for the application in the advertising field compared with the headset-based mixed reality system. Furthermore, if the above factors are not considered, this system can be deployed in the head set, as long as the headset supports third-party software and it meets the file format required by the headset.

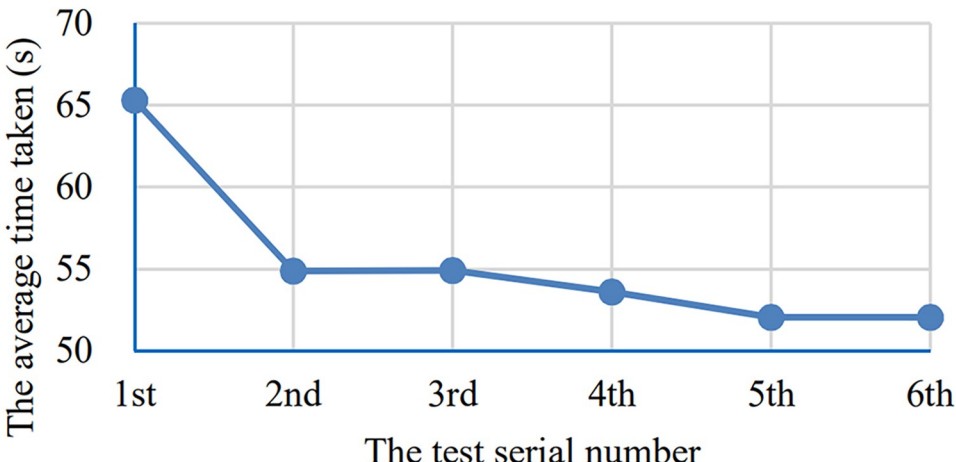

**Fig 15. The average time taken curve.** The horizontal axis indicates the test serial number and the vertical axis indicates the time taken of all the 22 participants in seconds.

**Table 3. The questionnaire survey content and the answers.**

| Index | Questions content | YES | NO | N/A |
|-------|-------------------|-----|-----|-----|
| Q1 | Do you think this system is more interesting compared with other forms of advertising? | 22 | 0 | 0 |
| Q2 | Will you be attracted by the interactive system and enter the store? | 15 | 4 | 3 |
| Q3 | Will you actively try to interact with the system if you know how to interact? | 22 | 0 | 0 |
| Q4 | Compared with other forms of advertising, do you have a deeper memory of the content displayed through the system? | 17 | 2 | 3 |
| Q5 | If a store (product) uses an interactive advertising system, will it increase your favoritism for the store (product)? | 20 | 1 | 1 |
| Q6 | If you are the person in charge of a store (product), will you choose this system for advertising (assuming that the advertising costs are the same)? | 19 | 2 | 1 |
| Q7 | Do you know how to operate the system through the operation manual? | 21 | 0 | 1 |
| Q8 | Do you think the gestures design of the system are appropriate? | 11 | 5 | 6 |
| Q9 | Do you think this system is easy to learn? | 21 | 0 | 1 |
| Q10 | Will you recommend this interactive system to your friends? | 21 | 0 | 1 |

## Conclusions

The advertising video system, designed based on augmented reality, creates a virtual cube on the marker of the frames collected by the camera, displaying the advertising videos on the faces of the cube. The plane of video playback is related to the pose of the marker collected by the camera instead of on the screen directly. The plane will change with the position of the marker in front of the camera in real time.

The support of gesture operation enables the advertising audience to freely choose the advertising content of interest without physical contact, which makes the advertising more interesting. In addition to its high efficiency and accuracy, it performs well in terms of learnability and potential applicability. With a lower cost compared to mixed reality, it can not only broadcast advertising videos, but also be applied to museums, zoos and other venues where multiple objects need to be explained.

This study still has limitations. Firstly, the markers do not include general images but black-and-white cells. For instance, if the icon of the advertising product is placed in the center of the mark, people may be more willing to scan it with their mobile phones rather than simple markers. Secondly, since up to six advertising videos can be played at the same time, their sounds will interfere with each other. As a result, the system does not support the playing of audio videos now. Furthermore, as suggested by the test participants, the system has some gestures that are not easy to make, and also lacks game factors which can attract customers' attention to a large extent. These limitations should be considered in our next work indubitably.

## Acknowledgments

We thank the reviewers for their helpful suggestions on the function enhancement and feature description of ARIAS.

## Author Contributions

**Formal analysis:** Qiujiao Wang.

**Funding acquisition:** Qiujiao Wang.

**Methodology:** Qiujiao Wang.

**Software:** Zhijie Xie.

**Visualization:** Zhijie Xie.

**Writing – original draft:** Zhijie Xie.

**Writing – review & editing:** Qiujiao Wang.

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
