## [Decision Letter · Decision Letter 0]

15 Feb 2023

PONE-D-22-35281

ARVS: Design of an interactive advertising system

PLOS ONE

Dear Dr. Wang,

Thank you for submitting your manuscript to PLOS ONE. After careful consideration, we feel that it has merit but does not fully meet PLOS ONE’s publication criteria as it currently stands. Therefore, we invite you to submit a revised version of the manuscript that addresses the points raised during the review process.

The reviewers have found minor issues in the manuscript. More specifically, the reviewer 1 asks for clarifications of terms used in the manuscript and more discussions regarding the comparison of the proposed system and the state of the art. The reviewer 2 has some concerns regarding the statistical analysis which was found not rigorous and appropriate. Moreover, the reviewer requests to define the acronym ARVS, provide clarifications regarding the use of the system and discuss the learning curve of using the proposed interaction method. 

We look forward to receiving your revised manuscript.

Kind regards,

Alberto Cannavò

Academic Editor

PLOS ONE

Journal Requirements:

7. We note that Figure 12 includes an image of a participant in the study.

8. We note you have included a table to which you do not refer in the text of your manuscript. Please ensure that you refer to Table 1 in your text; if accepted, production will need this reference to link the reader to the Table.

Additional Editor Comments:

After reading the paper and considering the reports by the reviewers, I think that the authors should implement the revisions requested in order for the paper to be considered for publication in the journal.

The reviewers have found minor issues in the manuscript. More specifically, the reviewer 1 asks for clarifications of terms used in the manuscript and more discussions regarding the comparison of the proposed system and the state of the art.

The reviewer 2 has some concerns regarding the statistical analysis which was found not rigorous and appropriate. Moreover, the reviewer requests to define the acronym ARVS, provide clarifications regarding the use of the system and discuss the learning curve of using the proposed interaction method.

Reviewers' comments:

Reviewer's Responses to Questions

**Comments to the Author**

1. Is the manuscript technically sound, and do the data support the conclusions?

Reviewer #1: Yes

Reviewer #2: Yes

2. Has the statistical analysis been performed appropriately and rigorously? 

Reviewer #1: N/A

Reviewer #2: No

3. Have the authors made all data underlying the findings in their manuscript fully available?

Reviewer #1: Yes

Reviewer #2: Yes

4. Is the manuscript presented in an intelligible fashion and written in standard English?

Reviewer #1: Yes

Reviewer #2: Yes

5. Review Comments to the Author

Reviewer #1: Abstact

Line 1: describe the meaning of ARVS

Authors could describe the Mixed Reality and make a comparison between this system and theirs

Authors could describe potential applicability of such system and the reason why it is different with the other systems.

Reviewer #2: The submission titled “ARVS: Design of an interactive advertising system” discusses a methodology to produce a virtual cube, akin to an augmented reality hologram, that shows advertising videos to better grasp customer attention. The authors use established external libraries that identify and transform individual video frames, and then implement hand gestures to incorporate human interactions. A user’s fingertips serve as a mouse pointer and hand gestures are used to translate, rotate, and zoom the virtual cube. Overall, this was a well-written manuscript, the data algorithms were adequately explained, and the significance of the work was properly highlighted from a psychological standpoint. I only have a few comments listed below.

1. The acronym ARVS should be defined. I assume it stands for augmented reality video system? Is this a novel name that the authors created?

2. Is there an indication given to the user for which state is currently selected (e.g., translation, rotation, zooming, etc.)? Otherwise, can virtual buttons be displayed to the user to facilitate interactions with the virtual cube?

3. How steep is the learning curve to use the gesture interactions? If there is a new customer that has not used ARVS, would they become too confused to benefit from the virtual cube advertising? There would be a great benefit to test the system on users that ranges of prior experience with ARVS.

4. More explanation is required to explain how gestures are tracked. Is there another camera directly across the user? Or could there be a camera placed behind the user? How adaptable is the system to augmented reality headsets that can similarly record hand gestures and/or gaze data?

6. PLOS authors have the option to publish the peer review history of their article (what does this mean?). If published, this will include your full peer review and any attached files.

Reviewer #1: No

Reviewer #2: No

---

## [Author Response · Author response to Decision Letter 0]

14 Mar 2023

Comments from the academic editor: 

The reviewers have found minor issues in the manu-script. More specifically, the reviewer 1 asks for clarifica-tions of terms used in the manuscript and more discus-sions regarding the comparison of the proposed system and the state of the art. The reviewer 2 has some con-cerns regarding the statistical analysis which was found not rigorous and appropriate. Moreover, the reviewer re-quests to define the acronym ARVS, provide clarifica-tions regarding the use of the system and discuss the learning curve of using the proposed interaction method.

Response:

Thank you very much. We have made corresponding modifications, as detailed below.

Comments from the Reviewer #1: 

Line 1: describe the meaning of ARVS

Response: 

Thank you very much for all of your helpful suggestions. 

We named the system 'A video system based on augmented reality' at first, and ARVS was its acronym. Since it has been upload to the web site of GitHub and open with a unique URL, we haven't changed its name later. Now we change it to ARIAS to avoid ambiguity and highlight its interactivity, and explained it in the title and abstract of the manuscript. (Line2-3, Line11-13)

Authors could describe the Mixed Reality and make a comparison between this system and theirs

Response: 

We discuss the advantages and disadvantages of AR in this system compared to MR, which with the similar theoretical technology with AR. The Similar sugges-tions are given by the other reviewer, and we added the ‘System features’ section for comprehensive descripting (Line567-603)

Authors could describe potential applicability of such system and the reason why it is different with the other systems.

Response: 

This article has four keyword: Advertising, Augmented reality, Gesture and In-teractive, we found none effective comparable system in GitHub and Web of Science with any three of them, including the use of their synonyms and acro-nyms. (Perhaps this system will become the compared object if it can be pub-lished). Therefore, this system was compared with the traditional way of adver-tising. We analysed the potential applicability of such system according to the results. (Line516-524, Line558-566)

Comments from the Reviewer #2: 

The submission titled “ARVS: Design of an interactive advertis-ing system” discusses a methodology to produce a virtual cube, akin to an augmented reality hologram, that shows advertising videos to better grasp customer attention. The authors use estab-lished external libraries that identify and transform individual video frames, and then implement hand gestures to incorporate human interactions. A user’s fingertips serve as a mouse pointer and hand gestures are used to translate, rotate, and zoom the vir-tual cube. Overall, this was a well-written manuscript, the data algorithms were adequately explained, and the significance of the work was properly highlighted from a psychological standpoint. I only have a few comments listed below.

Response: 

Thank you very much for praising our work and for all of your helpful suggestions. Your questions will be answered in detail below.

1. The acronym ARVS should be defined. I assume it stands for augmented reality video system? Is this a novel name that the authors created?

Response: 

Yes, we named the system 'An augmented reality video system' at first, and ARVS was its acronym. Since it has been upload to the web site of GitHub and open with a unique URL, we haven't changed its name, even though it has been added the function of gesture manipulation later. Now we change it to ARIAS to avoid ambiguity and highlight its interactivity, and explained it in the title and abstract of the manuscript. (Line2-3, Line11-13)

2. Is there an indication given to the user for which state is cur-rently selected (e.g., translation, rotation, zooming, etc.)? Oth-erwise, can virtual buttons be displayed to the user to facilitate interactions with the virtual cube?

Response: 

Thanks again for the helpful idea. We have always felt that these states and instructions are easy to remember from our own point of view, but ignored that it may be a challenge for a customer who has just contacted the system for a few minutes. Now, the virtual buttons and text indica-tion were both given to the user. (Line418-428,Fig 11)

3. How steep is the learning curve to use the gesture interac-tions? If there is a new customer that has not used ARVS, would they become too confused to benefit from the virtual cube adver-tising? There would be a great benefit to test the system on users that ranges of prior experience with ARVS.

Response: 

We invited 22 people who did not contacted the system to participate in the test of the system. Each person tested the system 6 times, and then obtained the number of errors and the time taken for each test. We drew the error curve and time taken curve, the results then were discussed according to the curve characteristics. (Line466-515, Line526-557)

4. More explanation is required to explain how gestures are tracked. Is there another camera directly across the user? Or could there be a camera placed behind the user? How adaptable is the system to augmented reality headsets that can similarly rec-ord hand gestures and/or gaze data?

Response: 

For the gesture tracking, it work with the help of the library MediaPipe, by which the gesture data is inferred from the image based on deep learning. We just mentioned briefly (Line297-303) because the working principle is described in detail in the 31th reference. So there is just only one camera required by this system. (Line593) 

For the system adaptable to headsets, the similar suggestions are given by the other reviewer, and we added the System features section for comprehensive descripting. (Line567-603)

Journal Requirements:

Response: 

We have checked it and corrected some errors.

Response: 

We have noticed the guidelines on code sharing and we think our code has met it.

Response: 

We are sorry for the wrong financial disclosure before. The authors received no specific funding for this work. Please help us change the online submission.

Response: 

We add the minimal data set to GitHub since the minimal data set is public as a part of this system. The URL has been declared in the manuscript.

Response: 

The two accounts have been associated.

Response: 

The ethics statement has been moved to the Methods section.

7. We note that Figure 12 includes an image of a participant in the study.

Response: 

We have got a written informed consent.

8. We note you have included a table to which you do not refer in the text of your manuscript. Please ensure that you refer to Table 1 in your text; if accepted, production will need this reference to link the reader to the Table.

Response: 

We have noticed this error and it has been revised.

Response: 

We checked the list and changed the style to 'Vancouver' and made some minor adjustments.

---

## [Decision Letter · Decision Letter 1]

11 Apr 2023

PONE-D-22-35281R1ARIAS: An AR-based interactive advertising systemPLOS ONE

Dear Dr. Wang,

Thank you for submitting your manuscript to PLOS ONE. After careful consideration, we feel that it has merit but does not fully meet PLOS ONE’s publication criteria as it currently stands. Therefore, we invite you to submit a revised version of the manuscript that addresses the points raised during the review process.

ACADEMIC EDITOR: All the issues have been addressed in the revised version of the paper. A final remark is raised by a reviewer that regards a figure in the manuscript.  Considering the reviewer's comment I recommend accepting the paper after a minor revision.

We look forward to receiving your revised manuscript.

Kind regards,

Alberto Cannavò

Academic Editor

PLOS ONE

Journal Requirements:

Additional Editor Comments (if provided):

According to the reviewers' comments, the majority of the concerns have been answered/addressed.

A final comment regards the format of Figure 10 that should be fixed.

Reviewers' comments:

Reviewer's Responses to Questions

**Comments to the Author**

1. If the authors have adequately addressed your comments raised in a previous round of review and you feel that this manuscript is now acceptable for publication, you may indicate that here to bypass the “Comments to the Author” section, enter your conflict of interest statement in the “Confidential to Editor” section, and submit your "Accept" recommendation.

Reviewer #1: All comments have been addressed

Reviewer #2: All comments have been addressed

2. Is the manuscript technically sound, and do the data support the conclusions?

Reviewer #1: Yes

Reviewer #2: Yes

3. Has the statistical analysis been performed appropriately and rigorously? 

Reviewer #1: Yes

Reviewer #2: Yes

4. Have the authors made all data underlying the findings in their manuscript fully available?

Reviewer #1: Yes

Reviewer #2: Yes

5. Is the manuscript presented in an intelligible fashion and written in standard English?

Reviewer #1: Yes

Reviewer #2: Yes

6. Review Comments to the Author

Reviewer #1: The Authors has replied to all comments satisfactorily and for this reason I suggest accepting the MS.

Reviewer #2: The authors have adequately answered/addressed my concerns. My final comment is to appropriately format Fig. 10 with axes labels.

7. PLOS authors have the option to publish the peer review history of their article (what does this mean?). If published, this will include your full peer review and any attached files.

Reviewer #1: No

Reviewer #2: No

---

## [Author Response · Author response to Decision Letter 1]

13 Apr 2023

Academic Editor:

All the issues have been addressed in the revised version of the paper. A final remark is raised by a reviewer that regards a figure in the manuscript. Considering the reviewer's comment I recommend accepting the paper after a minor revision. 

According to the reviewers' comments, the majority of the concerns have been answered/addressed. A final comment regards the format of Figure 10 that should be fixed.

Response:

Thank you very much for considering accepting our manuscript. The Fig.10 as well as Fig.12-15 has been formatted as the comment of Reviewer #2.

Reviewer #1:

The Authors has replied to all comments satisfactorily and for this reason I suggest accepting the MS.

Response:

Thank you very much for suggesting accepting our manuscript.

Reviewer #2:

The authors have adequately answered/addressed my concerns. My final comment is to appropriately format Fig. 10 with axes labels.

Response:

Thank you very much for your kind comment. The Fig.10 as well as Fig.12-15 has been formatted with axes labels.

---

## [Decision Letter · Decision Letter 2]

3 May 2023

ARIAS: An AR-based interactive advertising system

PONE-D-22-35281R2

Dear Dr. Wang,

We’re pleased to inform you that your manuscript has been judged scientifically suitable for publication and will be formally accepted for publication once it meets all outstanding technical requirements.

Kind regards,

Alberto Cannavò

Academic Editor

PLOS ONE

Additional Editor Comments (optional):

According to the comments of the reviewers, all the comments have been addressed in the revised version of the manuscript. For this reason, I suggest accepting the manuscript.

Reviewers' comments:

Reviewer's Responses to Questions

**Comments to the Author**

1. If the authors have adequately addressed your comments raised in a previous round of review and you feel that this manuscript is now acceptable for publication, you may indicate that here to bypass the “Comments to the Author” section, enter your conflict of interest statement in the “Confidential to Editor” section, and submit your "Accept" recommendation.

Reviewer #1: All comments have been addressed

Reviewer #2: All comments have been addressed

2. Is the manuscript technically sound, and do the data support the conclusions?

Reviewer #1: Yes

Reviewer #2: Yes

3. Has the statistical analysis been performed appropriately and rigorously? 

Reviewer #1: Yes

Reviewer #2: Yes

4. Have the authors made all data underlying the findings in their manuscript fully available?

Reviewer #1: Yes

Reviewer #2: Yes

5. Is the manuscript presented in an intelligible fashion and written in standard English?

Reviewer #1: Yes

Reviewer #2: Yes

6. Review Comments to the Author

Reviewer #1: Authors has replied to all comments satisfactorily. The Ms improved markedly and for this reason I suggest accepting it.

Reviewer #2: This version is sufficient for publication.

7. PLOS authors have the option to publish the peer review history of their article (what does this mean?). If published, this will include your full peer review and any attached files.

Reviewer #1: No

Reviewer #2: No

---

## [Editor Report · Acceptance letter]

2 Jun 2023

PONE-D-22-35281R2 

ARIAS: An AR-based interactive advertising system 

Dear Dr. Wang:

I'm pleased to inform you that your manuscript has been deemed suitable for publication in PLOS ONE. Congratulations! Your manuscript is now with our production department. 

Kind regards, 

on behalf of

Ph.D. Alberto Cannavò 

Academic Editor

PLOS ONE